# Common Frame Dynamics for Conically-Constrained Spacecraft Attitude Control

**DOI:** 10.3390/s222410003

**Published:** 2022-12-19

**Authors:** Arnold Christopher Cruz, Ahmad Bani Younes

**Affiliations:** Department of Aerospace Engineering, San Diego State University, San Diego, CA 92182-1308, USA

**Keywords:** attitude control, lyapunov control, common frame dynamics, constrained control

## Abstract

Attitude control subjected to pointing constraints is a requirement for most spacecraft missions carrying sensitive on-board equipment. Pointing constraints can be divided into two categories: exclusion zones that are defined for sensitive equipment such as telescopes or cameras that can be damaged from celestial objects, and inclusion zones that are defined for communication hardware and solar arrays. This work derives common frame dynamics that are fully derived for Modified Rodrigues Parameters and introduced to an existing novel technique for constrained spacecraft attitude control, which uses a kinematic steering law and servo sub-system. Lyapunov methods are used to redevelop the steering law and servo sub-system in the common frame for the tracking problem for both static and dynamic conic constraints. A numerical example and comparison between the original frame and the common frame for the static constrained tracking problem are presented under both unbounded and limited torque capabilities. Monte Carlo simulations are performed to validate the convergence of the constrained tracking problem for static conic constraints under small perturbations of the initial conditions. The performance of dynamic conic constraints in the tracking problem is addressed and a numerical example is presented. The result of using common frame dynamics in the constrained problem shows decreased control effort required to rotate the spacecraft.

## 1. Introduction

Spacecraft missions regularly require reorientation maneuvers subject to various constraints such as mechanical limitations and orientation constraints. Mechanical limitations are defined by maximum available torque for a reaction wheel array, or angular velocity limits of the spacecraft. Attitude constraint limits are defined for mission hardware which can be separated into two categories, inclusion zones or exclusion zones [1]. Inclusion zones are required for equipment that require a certain attitude to be considered for maximum performance such as solar panel arrays to pointing at the sun or communication antennas to point towards a ground station. Exclusion zones are required for sensitive equipment such as a telescope or camera boresight to be pointed away from the sun at a given angle to prevent severe sensor damage. Attitude pointing constraints are common for various spacecraft, from small cubesats to very expensive and complex satellites [2]. One notable example is the James Webb Space Telescope (JWST) which has an exclusion zone of $85^{\circ }$ from the sun and $45^{\circ }$ anti-sun [3]. An artist rendition is shown in Figure 1.

The autonomous attitude unconstrained problems, both in the constant reference regulation problem, and the moving reference tracking problem have been extensively researched. Some methods that have successfully solved the unconstrained problem are non-linear control [4,5], optimal control [4,6,7], sliding-mode control [8,9], and adaptive control [10,11]. The addition of pointing constraints to existing nonlinear kinematics and dynamics adds a challenge to solving the constrained attitude control problem since removing constrained zones from the feasible rotational configuration of the spacecraft creates nonconvex regions [12].

Hablani used geometric relations in order to determine trajectories that avoid constraint volume by defining an intermediate waypoint located outside the constraint [13]. From here, the spacecraft would maneuver to the final orientation. Although simple to implement, the algorithm does not scale with an increasing number of constraints [14]. Frakes et al. uses a different method that was implemented on the SAMPEX mission shown in Figure 2a that would avoid an orientation that would maximize the flux of orbital debris and micrometeoroids by monitoring the angle between the Heavy Ion Large Telescope (HILT) boresight and that condition [15]. If the boresight enters the conic constraint, a target attitude is redefined outside the constraint. This method is not guaranteed to converge.

Constraint Monitor Algorithms (CMT) are real-time algorithms that actively monitor constraints of all types and creates a trajectory avoiding them using a predictor-corrector approach [16]. In CMT, the current attitude motion is propagated forward for a short duration of time, and when the predicted motion violates the constraints corrective actions are taken. These algorithms have been originally designed for the Cassini mission [17] shown in Figure 2b and has also flown on the Deep-Space 1 mission. Although it is a sub-optimal solution, it guarantees the desired path and can be extended to a moving reference. However, CMT’s convergence is only guaranteed in some cases and not the general case [14].

Randomized motion planning has been used to solve the constrained problem [18], utilizing graph and random search to achieve the spacecrafts final orientation while avoiding constraints. The procedure is as follows [14]:Initialize a graph $G_{0}$ with a distinct vertex at $v_{0}$. This represents the initial states (attitude and angular velocity)At the k+1 iteration, perform a random graph search starting at the *k*th vertex $v_{k}$ to determine a set of feasible vertices in the graph $G_{k}$.Chose a feasible vertex found in the second step that minimizes the cost function. This is the next vertex $v_{k+1}$.From this vertex, repeat the second and third step from $v_{k+1}$. Update *k* to be k+1 until the final attitude is obtained.Apply the optimal control torque for each attitude trajectory.

The technique solves the constrained problem in the regulation problem; however, convergence is guaranteed only in probabilistic manner, and computational time increases dramatically as the graph size increases.

Optimal control methods can be used to solve the constrained problem, given a cost function that minimized control effort subjected to its initial state and terminal state bounded by spacecraft limitations. Although being a Non-Linear Optimization problem, Kim and Mesbahi developed a method to simplify the Non-Linear optimization Problem (NLP) to a Quadratically constrained quadratic programming problem (QCQP) and then solving the maneuver as a Semi-Definite Programming optimization problem by convexifying the constraint [16]. This method has been used to solve the constrained regulation problem and not the tracking problem. However, the method used by Walsh and Forbes solves both problems by transforming the constraint into the Direction Cosine Matrix (DCM) [1] representation and formulates the control problem using Semi-Definite Programming (SDP). This method is singularity free, as a characteristic of using the DCM, but is computationally expensive.

Lyapunov methods incorporate potential functions to create mathematically traceable control laws that converge to the target attitude while evading constraints and work with any number of conic constraints. Using these methods allows the use of backstepping control methods [19], allowing separate development of kinematics and angular rates. Ramos solved the constrained problem under bounded angular velocity by developing a kinematic steering law subject to a smoothing function and torque limitations by extending the constraint angle as a function of the spacecraft characteristics [20]. The technique conducted in reference [20] solves the regulation and tracking problem under an arbitrary number of constraints, but in the tracking problem the angular velocity is not guaranteed to be bounded; however, Ramos’s results present boundedness in angular velocity.

Adaptive optimal control methods have also been used. Kulumani and Lee [21] developed a geometric adaptive control system designed to asymptotically converge to the desired attitude while avoiding pointing constraints in presence of unknown disturbances using Special Orthogonal Group 3 (SO(3)) and various attitude representations. The constraints are also modeled using logarithmic barrier functions, modeled as attractive surfaces for inclusion zones and repulsive surfaces for exclusion zones and can be superimposed. The control system minimizes the negative gradient of the attitude error function. This method is singularity free and has been shown to solve both the tracking and regulation problem under an arbitrary number of constraints.

The key contributions of this paper are summarized below

The paper solves the constrained attitude control problem using common frame dynamics. Hence, a complete elegant constrained attitude control formulation in the common frame dynamics, where the angular velocity is defined in the estimated attitude axes frame. Conventionally, such as in the existing solution by Ramos [20], the problem does not consider the difference of frames between the body frame and the reference frame. It is required to use a different definition of the state error since a spacecraft’s Attitude Determination and Control System (ADCS) does not measure the attitude directly, but by using a set of angles to objects using sensors such as star cameras or sun sensors, while gyroscopes are used to measure angular rates [22]. More accurate sensors were developed during the space age and introduced new technology in Inertial Navigation Systems (INS) called the Inertial Measurement Units (IMUs), consisting of three gyroscopes and three accelerometers [23]. Despite the rise of improved INS sensors, IMUs are well known to drift. One example of this is the Apollo mission’s gyroscopes which drifted at a rate of one milliradian per hour [23]. In addition, since position measurements are only given, the calibration parameters of attitude and gyroscopes are weak since it is dependent on the spacecraft’s motion. In addition, the reference frame, which could be a solution from another spacecraft’s guidance algorithm, also contains the same type of errors as the body frame.The paper develops the common frame dynamics in the constrained attitude control problem. Common frame dynamics is introduced in previous work conducted by [20], which uses Modified Rodrigues Parameters (MRP’s) as minimal attitude descriptors. The backstepping control law is adopted to develop the kinematic steering laws and servo subsystem blocks, which simplify the design of the control laws by permitting the division of attitude and angular rates into separate control loops.The paper adopts the Lyapunov methods to develop mathematically traceable, closed-form control laws for both subsystems.In addition to implementing common frame dynamics, this paper also extends the constrained tracking problem in presence of dynamic constraints. These types of constraints are important in spacecraft formation flying, such as the European Data Relay System (EDRS) in Figure 3a and the upcoming Starlink satellite constellation with the purpose of providing low-latency satellite internet access globally (Further information can be seen at starlink.com (accessed on 1 March 2020).) in Figure 3b. Each satellite is equipped with a laser communication device and receiver with strict pointing inclusion constraints to transmit data across the constellation.The paper presents validation of the algorithm by performing a Monte Carlo analysis on two boresight trajectories under both exclusion constraints and explicitly under exclusion constraints. The constrained tracking problem is also examined in presence of dynamic constraints and exclusion constraints.

## 2. Common Frame Dynamics

Consider Figure 4, where N represents the Inertial Earth Frame with unit axes $\left[{\hat{i}}_{N},\;{\hat{j}}_{N},\;{\hat{k}}_{N}\right]$, B represents the moving body frame with axes $\left[{\hat{i}}_{B},\;{\hat{j}}_{B},\;{\hat{k}}_{B}\right]$, and R represents the reference frame with axes $\left[{\hat{i}}_{R},\;{\hat{j}}_{R},\;{\hat{k}}_{R}\right]$.

The angular velocity of the body frame is represented as ${}^{B}\omega$, while the reference angular velocity is represented as ${}^{R}\omega$. If the attitude is represented as σ in the body frame relative to the inertial frame (noted as $\sigma_{B/N}$), then ω is the angular velocity of the body frame relative to the inertial frame written in the body frame coordinate system (noted as ${}^{B}\omega_{B/N}$). For derivatives, the over dot symbol $\dot{•}$ represents the inertial derivative, while the prime symbol ${•}^{\prime }$ represents the body frame or reference frame derivative.

In Figure 5, let ${}^{B}C$ represent the Direction Cosine Matrix (DCM) in the body frame attitude with axes ${[}^{B}x{\;}^{B}y{\;}^{B}z]$ and let ${}^{R}C$ be the DCM in the reference frame attitude with axes ${[}^{R}x{\;}^{R}y{\;}^{R}z]$.

Bani Younes and Mortari [24,25] described the attitude error as the rotation error between the both frames:(1)$$\delta C={\;}^{B}C^{R}C^{T}$$

If the product between δC and the reference attitude ${}^{R}C$ is taken, then the result is the attitude in the body frame:$$\delta C^{R}C={{(}^{B}C^{R}C^{T})}^{R}C={\;}^{B}C$$

Therefore, the matrix δC in Equation (Equation 1) is the transformation matrix from the reference frame to the body frame.

Similarly, let ${}^{B}\omega$ represent the angular velocity vector of the body frame with the same axes as ${}^{B}C$ and let ${}^{R}\omega$ represent the angular velocity of the reference plane with the same axes as ${}^{R}C$. The angular velocity error can be defined as:(2)$$\delta \omega ={\;}^{B}\omega -{\;}^{R}\omega$$

The dynamics for Equation (Equation 2) is:(3)$$\delta \dot{\omega }={\;}^{B}\dot{\omega }-{\;}^{R}\dot{\omega }$$

If the body attitude is known, then Equation (Equation 2) represents the angular velocity error between the body and the reference frame. Equation (Equation 2) is used in simulation cases where the body attitude is known. In real applications, the body attitude is not known and the reference frame angular velocity must be transformed into the common frame. The Common Frame angular velocity error can be defined as:(4)$$\delta \omega ={\;}^{B}\omega -\delta C^{R}\omega$$

The dynamics for Equation Equation 4 is:(5)$$\delta \dot{\omega }={\;}^{B}\dot{\omega }-\delta {\dot{C}}^{R}\omega -\delta C^{R}\dot{\omega }$$
where $\delta \dot{C}=-\delta C\left[\delta \tilde{\omega }\right]$ and $\left[\delta \tilde{\omega }\right]$ is the skew symmetric matrix. For Equation (Equation 4), δC used as the corrective rotational matrix that maps the reference frame into the body frame.

The Modified Rodrigues Parameters (MRP) are a minimal attitude parameter set of Special Orthogonal Group 3 SO(3) and is defined in terms of the quaternions or in the principal rotation set [24,25,26]:(6)$$\sigma =\frac{q_{v}}{1+q_{4}}=e\tan\left(\frac{\phi }{4}\right)$$
where $q_{v}$ is the vector portion of the quaternion and $q_{4}$ is the scalar part of the quaternion.

Some properties of the MRPs are the geometric singularity and being a non-unique attitude representation. The geometric singularity is located at a principal angle of $\phi =\pm 360^{\circ }$, correlating to $q_{4}=-1$, allowing for large rotations. The non-unique character of the MRPs is expected, since it is a geometric projection of the quaternions, which are also non-unique since q=−q and as a result, a corresponding shadow set, $\sigma^{S}$ also represents the same orientation:(7)$$\sigma^{S}=-\frac{\sigma }{\sigma^{T}\sigma }$$

The MRP inverse transformation to the quaternions are given by:(8)$$q_{v}=\frac{2\sigma }{1+\sigma^{2}}\;\;\;\mathrm{and}\;\;\;q_{4}=\frac{1-\sigma^{2}}{1+\sigma^{2}}$$
and the DCM mapping of the MRP is:(9)$$\left[C\right]=\left[I_{3\times 3}\right]+\frac{8{\left[\tilde{\sigma }\right]}^{2}-4\left(1-\sigma^{2}\right)\left[\tilde{\sigma }\right]}{{\left(1+\sigma^{2}\right)}^{2}}$$
where $\left[I_{3\times 3}\right]$ is the identity matrix, and $\left[\tilde{\sigma }\right]$ is the skew symmetric matrix of the MRP defined as:$$\left[\tilde{\sigma }\right]=\left[\begin{array}{ccc} 0 & -\sigma_{3} & \sigma_{2}\\ \sigma_{3} & 0 & -\sigma_{1}\\ -\sigma_{2} & \sigma_{1} & 0 \end{array}\right]$$

To calculate the MRP Kinematic equation in the simulation case in Equation (Equation 2), the time derivative of Equation (Equation 6) is taken:(10)$$\dot{\sigma }=\frac{{\dot{q}}_{v}}{1+q_{4}}-\frac{{\dot{q}}_{4}q_{v}}{{\left(1+q_{4}\right)}^{2}}$$
and the corresponding derivatives for the quaternion components are:(11)$$\begin{array}{cc} {\dot{q}}_{v} & =\frac{1}{2}\left\{-\left({[}^{B}\tilde{\omega }]+2{[}^{R}\tilde{\omega }]\right)q_{v}+q_{4}{\;}^{B}\omega \right\}\\ {\dot{q}}_{4} & =-\frac{1}{2}{\;}^{B}\omega^{T}q_{v} \end{array}$$

Combining of Equation (Equation 10) and the quaternion kinematics in Equation (Equation 11) results in the MRP Kinematic differential equation when the true attitude and angular velocity are known [24,25]:(12)$$\dot{\sigma }=\frac{1}{4}\left[-2\left({[}^{B}\tilde{\omega }]+2{[}^{R}\tilde{\omega }]\right)\sigma +\left(1+\sigma^{2}\right){\;}^{B}\omega \right]+\frac{1}{2}\left({}^{B}\omega^{T}\sigma \right)\sigma$$
where $\left[\tilde{\omega }\right]$ is the skew symmetric matrix in the form of:$$\left[\tilde{\omega }\right]=\left[\begin{array}{ccc} 0 & -\omega_{3} & \omega_{2}\\ \omega_{3} & 0 & -\omega_{1}\\ -\omega_{2} & \omega_{1} & 0 \end{array}\right]$$

To obtain the MRP Kinematic differential equation in the Common Frame case presented in Equation (Equation 4), the time derivative presented in Equation (Equation 10) is used and the corresponding quaternion kinematics are taken.
(13)$$\begin{array}{cc} {\dot{q}}_{v} & =\frac{1}{2}\left\{-{[}^{B}\tilde{\omega }]q_{v}+\omega q_{4}\right\}\\ {\dot{q}}_{4} & =-\frac{1}{2}{\;}^{B}\omega^{T}q_{v} \end{array}$$

Using the inverse mapping of the Common Frame quaternion kinematics presented in Equation (Equation 8), the MRP Kinematic Equation in the Common Frame simplifies to:(14)$$\dot{\sigma }=\frac{1}{4}\left[\left(1-\sigma^{2}\right)I_{3\times 3}+2{[}^{B}\tilde{\omega }]+2\sigma^{T}\sigma \right]{\;}^{B}\omega$$

Equations (Equation 2) and (Equation 12) represent the exact analytical solution for the true frame and Equations (Equation 4) and (Equation 14) represent the exact common frame analytical solution by [24]. In [27,28,29,30], the kinematic differential equation is derived for multiple attitude representations such as quaternions, Classical Rodrigues Parameters (CRP), and Direction Cosine Matrix (DCM) in the true and the common frame scenario.

The rotational equations of motion for a rigid spacecraft with $N_{RW}$ perfectly symmetric and balanced Reaction Wheels (RW) is given by [26]:(15)$$\left[I_{RW}\right]\dot{\omega }=-\left[\tilde{\omega }\right]\left(\left[I_{RW}\right]\omega +\left[G_{s}\right]h_{s}\right)-\left[G_{s}\right]u_{s}+L$$
where $\left[I_{RW}\right]$ is the spacecraft total inertia tensor with the RW system:(16)$$\left[I_{RW}\right]=\left[I_{s}\right]+\sum_{i=1}^{N_{RW}}\left(J_{t_{i}}{\hat{g}}_{t_{i}}^{T}{\hat{g}}_{t_{i}}+J_{g_{i}}{\hat{g}}_{g_{i}}^{T}{\hat{g}}_{g_{i}}\right)$$

$\left[G_{s}\right]$ is the RW projection matrix with respect to the spin axis:(17)$$\left[G_{s}\right]=\left[{\hat{g}}_{s_{1}},{\hat{g}}_{s_{i}},\ldots ,{\hat{g}}_{s_{N_{RW}}}\right]$$

$h_{s}$ is the RW angular momentum vector:(18)$$h_{s}={\left[J_{s_{1}}\left({\hat{g}}_{s_{1}}^{T}\omega +\Omega_{1}\right)\ldots J_{s_{i}}\left({\hat{g}}_{s_{i}}^{T}\omega +\Omega_{i}\right)\ldots J_{s_{N_{RW}}}\left({\hat{g}}_{s_{N_{RW}}}^{T}\omega +\Omega_{N_{RW}}\right)\right]}^{T}$$

$\left[I_{s}\right]$ is the inertia tensor of the spacecraft. A principal-axis frame $W_{i}:\left\{{\hat{g}}_{s_{i}},{\hat{g}}_{t_{i}},{\hat{g}}_{g_{i}}\right\}$ is applied to each RW. $\left[I_{w_{i}}\right]=diag\left(J_{s_{i}},J_{t_{i}},J_{g_{i}}\right)$ is the inertia of each RW relative to the center of mass within the W. $\Omega_{i}$ is the angular velocity of the RW relative to the spacecraft. The vector $u_{s}$ is the control torque vector applied to each RW axis. L is the external torque applied to the spacecraft.

## 3. Common Frame Control

The unconstrained tracking problem can be solved using Lyapunov’s direct method which allows similar methods in developed in backstepping control. This results in producing a cascaded control design where the output of one sub-system is the input of another [19]. Figure 6 describes the control design used in [20]. The kinematics block represents Equation (Equation 14), the MRP kinematic differential equation in the Common Frame. The dynamics block is represented by Equation (Equation 15), the rigid body dynamics of a spacecraft with a RW array. The outer loop of the control block is the Kinematic Steering Law, responsible for controlling the attitude of the spacecraft and taking inputs from the reference $\sigma_{R/N}$, and the current attitude $\sigma_{B/N}$. The inner loop is the servo sub-system, responsible for controlling the angular velocity and takes inputs from the reference $\omega_{R/N}$, the solution of the kinematic steering block $\omega_{B*/R}$ and the current angular velocity from the dynamics block $\omega_{B/N}$.

### 3.1. Unconstrained Kinematic Steering Law

Consider the Lyapunov candidate function given by [26]:(19)$$V\left(\sigma_{B/R}\right)=2\ln\left(1+\sigma_{B/R}^{T}\sigma_{B/R}\right)$$

Taking the time derivative:(20)$$\dot{V}\left(\sigma_{B/R}\right)=\frac{4\sigma_{B/R}^{T}{\dot{\sigma }}_{B/R}}{1+\sigma_{B/R}^{T}\sigma_{B/R}}$$

Using Equation (Equation 14) and knowing that $\sigma_{B/R}^{T}\left[{\tilde{\sigma }}_{B/R}\right]=0$, Equation (Equation 20) can be rewritten as:(21)$$\dot{V}\left(\sigma_{B/R}\right)=\sigma_{B/R}^{T}{\;}^{B}\omega_{B/R}$$

In order for the system to be asymptotically stable, Equation (Equation 21) must be negative definite. Let $\omega_{B^{*}/R}$ be the desired angular velocity of the body frame, $B^{*}$ relative to the reference frame, R. The steering law command is represented as:(22)$${}^{B}\omega_{B^{*}/R}=-f\left(\sigma_{B/R}\right)$$
where $f\left(\sigma_{B/R}\right)$ is an even function [31] such that:(23)$$\sigma^{T}f\left(\sigma \right)>0$$

Substituting Equation (Equation 22) into Equation (Equation 21):(24)$$\dot{V}=-\sigma_{B/R}^{T}f\left(\sigma_{B/R}\right)<0$$

Equation (Equation 24) is represents a general steering law expression in which $f(\sigma_{B/R})$ can be any even function and guarantees global asymptotic stability.

Since the MRP shadow set parameters in Equation (Equation 7) are being used to avoid the MRP singularity at $\pm 360^{\circ }$, then $\left|\sigma_{B/R}\right|$ upper bound is limited by 1. $f(\sigma_{B/R})$ is modified to control how fast the commanded rates approach $\omega_{max}$ by making $f(\sigma_{B/R})$ an odd function with the inclusion of a cubic term [26,31]:(25)$$f\left(\sigma \right)=f_{i}\left(\sigma_{i}\right)=\frac{2\omega_{max}}{\pi }\arctan\left(\left[K_{1}\sigma_{i}+K_{3}\sigma_{i}^{3}\right]\frac{\pi }{2\omega_{max}}\right)$$

### 3.2. Servo-Sub System

In order to track the desired body angular velocity commands, a servo sub-system is necessary to produce the required torques. The angular velocity tracking error is defined as [26,31]:(26)$$\omega_{B^{*}/B}=\omega_{B/N}-\omega_{B^{*}/N}$$
where $\omega_{B^{*}/N}$ in the common frame is defined as:(27)$$\omega_{B^{*}/N}=\omega_{B^{*}/R}+\delta C\omega_{R^{/}N}$$
where $\delta C={\;}^{B}C^{R}C^{T}$ and remaps the reference frame coordinates system to the body frame coordinate system. To create a rate-servo robust to unmodeled torque [26], an integral term must be added. The integral state z is defined as:(28)$$z=\int_{t_{0}}^{t_{f}}{\;}^{B}\omega_{B^{*}/B}\;d\tau$$

Now consider the rate servo Lyapunov candidate function [26,31]:(29)$$V_{\omega }\left(\omega_{B^{*}/B},z\right)=\frac{1}{2}\omega_{B^{*}/B}^{T}\left[I_{RW}\right]\omega_{B^{*}/B}+\frac{1}{2}z^{T}\left[K_{I}\right]z$$

The required torques for each RW can be found by moving $\left[G_{s}\right]u_{s}$ to the left hand side of the equation:(30)$$\left[G_{s}\right]u_{s}=L_{r}$$
where $L_{r}$ represents the right hand side terms:(31)$$\begin{array}{cc} L_{r}= & \left[P\right]\omega_{B^{*}/B}+\left[K_{I}\right]z+L-\left[{\tilde{\omega }}_{B/N}\right]\left(\left[I_{RW}\right]\omega_{B/N}+\left[G_{s}\right]h_{s}\right)+\\ & \left[I_{RW}\right]\left[{\tilde{\omega }}_{B/R}\right]\delta C\omega_{R/N}-\left[I_{RW}\right]\left(\omega_{B^{*}/R}^{\prime }+\delta C\left({\dot{\omega }}_{R/N}-\omega_{B/N}\times \omega_{R/N}\right)\right) \end{array}$$

For the general case of a redundant set of RW present, the minimum norm inverse is used to map the motor torques [26]:(32)$$u_{s}={\left[G_{s}\right]}^{T}{\left(\left[G_{s}\right]{\left[G_{s}\right]}^{T}\right)}^{-1}L_{r}$$

A full derivation for the right-hand side terms and numerical derivatives are discussed in [32].

## 4. Attitude Constrained Maneuver

Attitude pointing constraints that often appear in spacecraft missions are exclusion and inclusion constraints, which is further classified into four different categories [14]. Type I constraints are static hard constraints, defined by relatively stationary celestial objects with respect to the inertial frame. These are strict exposed or non-exposed constraints defined by sensitive equipment. Type II constraints are relaxations of Type I constraints that allow some violations. Type III constraints are dynamic constraints in which the constraint is time dependent. Type IV constraints are a combination of Type I to Type III constraints.

### 4.1. Static Conic Exclusion and Inclusion Constraints

Figure 7 is a diagram for the static exclusion and inclusion constraint. For the static exclusion constraint, the objective is to maneuver the spacecraft with sensitive equipment in the body frame defined by the unit vector $\hat{b}$ while avoiding the exclusion cone defined by a celestial pointing unit vector $\hat{n}$, with a minimum security angle of $\theta_{min}$.

Since the boresight vector and the celestial pointing vector are in different reference frames, the inertial vector must be transformed to the body matrix. For an exclusion constraint, the static constraint is defined as:(33)$$C_{\left[BN\right]}^{E}={\;}^{N}{\hat{n}}^{T}{\left[BN\right]}^{TB}\hat{b}-\cos\left(\theta_{min}\right)<0$$

In a similar way for the static inclusion constraint, the objective is to maneuver the spacecraft with equipment in the body frame defined by the unit vector $\hat{b}$ inside an inclusion cone defined by a celestial pointing unit vector $\hat{n}$ and security angle of $\theta_{min}$. Mathematically, an inclusion constraint is described as:(34)$$C_{\left[BN\right]}^{I}={\;}^{N}{\hat{n}}^{T}{\left[BN\right]}^{TB}\hat{b}-\cos\left(\theta_{min}\right)>0$$

Since inclusion and exclusion constraints use the same conic formulation through a function $C_{\left[BN\right]}\left([BN]\right)$, the following inequality constraint must be true [20]:(35)$$-2\leq C_{\left[BN\right]}\left([BN]\right)\leq 2$$

${\dot{C}}_{\left[BN\right]}\left([BN]\right)$ is computed using the transport theorem and the circular shift property of the triple product [26]. Taking the derivatives in the inertial fame, assuming that $\hat{n}$ is inertially constant and $\hat{b}$ is body-fixed, then:$${\dot{C}}_{\left[BN\right]}\left([BN]\right)=\frac{{}^{N}d\hat{n}}{dt}\cdot \hat{b}+\hat{n}\cdot \frac{{}^{N}d\hat{b}}{dt}=\hat{n}\cdot \left(\omega_{B/N}\times \hat{b}\right)$$

Applying the circular shift property of the triple product:$$\begin{array}{cc} \hat{n}\cdot \left(\omega_{B/N}\times \hat{b}\right)= & \left(\hat{b}\times \hat{n}\right)\cdot \omega_{B/N} \end{array}$$

Finally, ${\dot{C}}_{\left[BN\right]}\left([BN]\right)$ becomes
(36)$${\dot{C}}_{\left[BN\right]}\left([BN]\right)={\left(\left[{}^{B}\tilde{b}\right]{\left[BN\right]}^{N}\hat{n}\right)}^{TB}\omega_{B/N}$$

Equations (Equation 33), (Equation 34) and (Equation 36), can be written in terms of the MRPs by transforming $\sigma_{B/N}$ to the DCM. These equations can be rewritten as:(37)$$C_{\left[BN\right]}\left(\left[BN\right]\left(\sigma_{B/N}\right)\right)={\;}^{N}{\hat{n}}^{T}{\left[BN\left(\sigma_{B/N}\right)\right]}^{TB}\hat{b}-\cos\left(\theta_{min}\right)$$
(38)$${\dot{C}}_{\left[BN\right]}\left(\left[BN\right]\left(\sigma_{B/N}\right)\right)={\left(\left[{}^{B}\tilde{b}\right]{\left[BN\left(\sigma_{B/N}\right)\right]}^{N}\hat{n}\right)}^{TB}\omega_{B/N}$$

### 4.2. Dynamic Conic Exclusion and Inclusion Constraints

In Figure 8, a dynamic conic constraint is shown. For a dynamic exclusion constraint, the goal is to maneuver the spacecraft while avoiding a body-fixed unit vector $\hat{b}$ from entering the exclusion cone defined by a moving unit vector $\hat{n}\left(t\right)$, and a security angle $\theta_{min}$:(39)$$C_{\left[BN\right]}^{E}={\;}^{N}\hat{n}{\left(t\right)}^{T}{\left[BN\left(\sigma_{B/N}\right)\right]}^{TB}\hat{b}-\cos\left(\theta_{min}\right)<0$$

In a similar way, the goal for a dynamic inclusion constraint is to keep the boresight vector $\hat{b}$ inside an inclusion cone defined by a moving unit vector $\hat{n}\left(t\right)$ and security angle $\theta_{min}$:(40)$$C_{\left[BN\right]}^{I}={\;}^{N}\hat{n}{\left(t\right)}^{T}{\left[BN\left(\sigma_{B/N}\right)\right]}^{TB}\hat{b}-\cos\left(\theta_{min}\right)>0$$

The inequality condition presented by Equation (Equation 35) also holds true. The derivative of the dynamic constraint, ${\dot{C}}_{\left[BN\right]}\left([BN]\right)$ is computed in a similar way shown in Section 4.1. ${\dot{C}}_{BN}\left(\left[BN\right]\right)$ becomes:(41)$${\dot{C}}_{\left[BN\right]}\left(\left[BN\right]\left(\sigma_{B/N}\right)\right)=\frac{{}^{N}d\hat{n}\left(t\right)}{dt}\cdot \hat{b}+{\left(\left[{}^{B}\tilde{b}\right]{\left[BN\left(\sigma_{B/N}\right)\right]}^{N}\hat{n}\right)}^{TB}\omega_{B/N}$$

The first term becomes:$${\;}^{N}\frac{d\hat{n}}{dt}\cdot \hat{b}={\;}^{N}\dot{\hat{n}}=\frac{d\hat{n}}{dt}{\left[BN\right]}^{T}\hat{b}$$

Therefore, ${\dot{C}}_{BN}\left(\left[BN\right]\right)$ for the dynamic constraint becomes:(42)$${\dot{C}}_{\left[BN\right]}\left(\left[BN\right]\left(\sigma_{B/N}\right)\right)={\;}^{N}\dot{\hat{n}}{\left(t\right)}^{T}{\left[BN\left(\sigma_{B/N}\right)\right]}^{T}\hat{b}+{\left(\left[{}^{B}\tilde{b}\right]{\left[BN\left(\sigma_{B/N}\right)\right]}^{N}\hat{n}\right)}^{TB}\omega_{B/N}$$

### 4.3. Constrained Attitude Control

For the constrained problem, consider $N_{E}$, the number of exclusion zones defined by $C_{i}^{E}:SO\left(3\right)\overset{}{\to }R$ and $N_{j}^{I}:SO\left(3\right)\overset{}{\to }R$, which can be functions described by Equation (Equation 37). Let D be a feasible attitude set such that [20]:$$D=\left\{x\in SO\left(3\right):\left[C_{i}^{E}\left(x\right)<0\;\wedge \;C_{j}^{I}\left(x\right)>0\right]\right\}$$

The objective of the controller is to drive the attitude error $\sigma_{B/R}$ and the angular velocity error $\omega_{B/R}$ to zero while moving in D. The first necessary condition is:$$\left[BN\right]\in D\;\forall \;t=\left[0,t_{f}\right]$$

To design control laws using Lyapunov’s direct method that considers attitude constraints, logarithmic barrier functions are used to convexify the constraints, resulting in smooth and strictly convex control functions [12].

### 4.4. Constrained Kinematic Steering Law Design

The goal of the tracking problem is to steer the spacecraft such that $\sigma_{B/R}\overset{}{\to }0$ and $\omega_{B/R}\overset{}{\to }0$. Similar to the unconstrained laws in Section 3.1, a servo sub-system and a steering law is required to control the angular velocity and attitude, respectively. The servo sub-system does not change; however, the steering law must be modified to consider the attitude constraints.

The Lyapunov candidate function becomes, $V:D\overset{}{\to }R^{+}$ [20]:(43)$$\begin{array}{cc} V\left(\sigma_{B/R}\right)=\; & 2\ln\left(1+\sigma_{B/R}^{T}\sigma_{B/R}\right)\\ \; & \left[-\frac{1}{N_{E}}\sum_{i=1}^{N_{E}}\ln\left(-\frac{C_{i}^{E}\left(\sigma_{B/N}\right)}{\alpha_{i}}\right)-\frac{1}{N_{I}}\sum_{j=1}^{N_{I}}\ln\left(\frac{C_{j}^{I}\left(\sigma_{B/N}\right)}{\beta_{j}}\right)\right] \end{array}$$

Compared to the unconstrained counterpart in Equation (Equation 43), the Lyapunov function depends on $\sigma_{B/R}$ and $\sigma_{B/N}$. Note that the Lyapunov function, Equation (Equation 43), is split into two parts:unconstrained, which is given by:
$$\begin{array}{c} V_{uncon}=2\ln\left(1+\sigma_{B/R}^{T}\sigma_{B/R}\right) \end{array}$$constrained, which is given by:
$$\begin{array}{c} V_{con}=\left[-\frac{1}{N_{E}}\sum_{i=1}^{N_{E}}\ln\left(-\frac{C_{i}^{E}\left(\sigma_{B/N}\right)}{\alpha_{i}}\right)-\frac{1}{N_{I}}\sum_{j=1}^{N_{I}}\ln\left(\frac{C_{j}^{I}\left(\sigma_{B/N}\right)}{\beta_{j}}\right)\right] \end{array}$$The parameters $\alpha_{i}>0$ and $\beta_{j}>0$ are chosen such that:
(44)$$\left[-C_{i}^{E}\left(\sigma_{B/N}\right)<\alpha_{i},C_{j}^{I}\left(\sigma_{B/N}\right)<\beta_{i}\right]\forall \sigma_{B/N}\in D$$

One possibility arises from the inequality constraint imposed by Equation (Equation 35). As a result, one choice is for $\alpha_{i}=\beta_{j}=2e$. Then the logarithmic constraints to be bounded between 1 and +∞. Equation (Equation 43) has the following characteristics [20]:V is continuously differential in DV(0)=0$V(\sigma_{B/R})>0$$\forall \;\sigma_{B/R}\in \left\{D-\left\{0\right\}\right\}$If the parameters are chosen such that $\alpha_{i}>0$ and $\beta_{j}>0$.The conditions in Equation (Equation 44) are satisfied. Since Equation (Equation 44) and −ln(x) are strictly decreasing functions:
$$\left[-\frac{1}{N_{E}}\sum_{i=1}^{N_{E}}\ln\left(-\frac{C_{i}^{E}\left(\sigma_{B/N}\right)}{\alpha_{i}}\right)-\frac{1}{N_{I}}\sum_{j=1}^{N_{I}}\ln\left(\frac{C_{j}^{I}\left(\sigma_{B/N}\right)}{\beta_{j}}\right)\right]>-\ln\left(1\right)-\ln\left(1\right)=0$$Given that $\ln\left(1+\sigma_{B/R}^{T}\sigma_{B/R}\right)>0\;\forall \sigma_{B/R}\in \left\{D-\left\{0\right\}\right\}$. It can be concluded that $V(\sigma_{B/R})>0$$\forall \;\sigma_{B/R}\in \left\{D-\left\{0\right\}\right\}$.For $\alpha_{i}=\beta_{j}=2e$:
$$\left[-\frac{1}{N_{E}}\sum_{i=1}^{N_{E}}\ln\left(-\frac{C_{i}^{E}\left(\sigma_{B/N}\right)}{\alpha_{i}}\right)-\frac{1}{N_{I}}\sum_{j=1}^{N_{I}}\ln\left(\frac{C_{j}^{I}\left(\sigma_{B/N}\right)}{\beta_{j}}\right)\right]>-\ln\left(\frac{1}{e}\right)-\ln\left(\frac{1}{e}\right)=2$$$V\left(\sigma_{B/R}\right)\overset{}{\to }+\infty$ when either $C_{i}^{E}\overset{}{\to }0$ or $C_{j}^{I}\overset{}{\to }0$

**Proof.** As $C_{i}^{E}\overset{}{\to }0$ or $C_{j}^{I}\overset{}{\to }0$: $-\ln\left(x\right)\overset{}{\to }+\infty$ and since $\ln\left(1+\sigma_{B/R}^{T}\sigma_{B/R}\right)\overset{}{\to }+\infty$, then it follows that $V\left(\sigma_{B/R}\right)\overset{}{\to }+\infty$ □

By conditions (1), (2), and (3), Equation (Equation 43) is a proper Lyapunov Function, bounded by domain D by condition (5). The time derivative of Equation (Equation 43) is given by: (45)$$\begin{array}{cc} \dot{V}\left(\sigma_{B/R}\right)=\; & \frac{4\sigma_{B/R}^{T}{\dot{\sigma }}_{B/R}}{1+\sigma_{B/R}^{T}\sigma_{B/R}}\left[-\frac{1}{N_{E}}\sum_{i=1}^{N_{E}}\ln\left(-\frac{C_{i}^{E}\left(\sigma_{B/N}\right)}{\alpha_{i}}\right)-\frac{1}{N_{I}}\sum_{j=1}^{N_{I}}\ln\left(\frac{C_{j}^{I}\left(\sigma_{B/N}\right)}{\beta_{j}}\right)\right]\\ \; & +2\ln\left(1+\sigma_{B/R}^{T}\sigma_{B/R}\right)\left[-\frac{1}{N_{E}}\sum_{i=1}^{N_{E}}\frac{{\dot{C}}_{i}^{E}\left(\sigma_{B/N}\right)}{C_{i}^{E}\left(\sigma_{B/N}\right)}-\frac{1}{N_{I}}\sum_{j=1}^{N_{I}}\frac{{\dot{C}}_{j}^{I}\left(\sigma_{B/N}\right)}{C_{j}^{I}\left(\sigma_{B/N}\right)}\right] \end{array}$$

Using Equation (Equation 14) and knowing that $\sigma_{B/R}^{T}\left[{\tilde{\sigma }}_{B/R}\right]=0$, Equation (Equation 45) can be rewritten as:(46)$$\begin{array}{cc} \dot{V}\left(\sigma_{B/R}\right)=\; & \sigma_{B/R}^{T}{\;}^{B}\omega_{B/R}\left[-\frac{1}{N_{E}}\sum_{i=1}^{N_{E}}\ln\left(-\frac{C_{i}^{E}\left(\sigma_{B/N}\right)}{\alpha_{i}}\right)-\frac{1}{N_{I}}\sum_{j=1}^{N_{I}}\ln\left(\frac{C_{j}^{I}\left(\sigma_{B/N}\right)}{\beta_{j}}\right)\right]\\ \; & +2\ln\left(1+\sigma_{B/R}^{T}\sigma_{B/R}\right)\left[-\frac{1}{N_{E}}\sum_{i=1}^{N_{E}}\frac{{\dot{C}}_{i}^{E}\left(\sigma_{B/N}\right)}{C_{i}^{E}\left(\sigma_{B/N}\right)}-\frac{1}{N_{I}}\sum_{j=1}^{N_{I}}\frac{{\dot{C}}_{j}^{I}\left(\sigma_{B/N}\right)}{C_{j}^{I}\left(\sigma_{B/N}\right)}\right] \end{array}$$

Substituting Equation (Equation 38) into Equation (Equation 46): (47)$$\begin{array}{cc} \; & \dot{V}\left(\sigma_{B/R}\right)=\;\sigma_{B/R}^{T}{\;}^{B}\omega_{B/R}\left[-\frac{1}{N_{E}}\sum_{i=1}^{N_{E}}\ln\left(-\frac{C_{i}^{E}\left(\sigma_{B/N}\right)}{\alpha_{i}}\right)-\frac{1}{N_{I}}\sum_{j=1}^{N_{I}}\ln\left(\frac{C_{j}^{I}\left(\sigma_{B/N}\right)}{\beta_{j}}\right)\right]\\ \; & +2\ln\left(1+\sigma_{B/R}^{T}\sigma_{B/R}\right)\left[-\frac{1}{N_{E}}\sum_{i=1}^{N_{E}}\frac{{\left({[}^{B}\tilde{b}]{\left[BN\right]}^{N}\hat{n}\right)}^{T}}{C_{i}^{E}\left(\sigma_{B/N}\right)}-\frac{1}{N_{I}}\sum_{j=1}^{N_{I}}\frac{{\left({[}^{B}\tilde{b}]{\left[BN\right]}^{N}\hat{n}\right)}^{T}}{C_{j}^{I}\left(\sigma_{B/N}\right)}\right]{\;}^{B}\omega_{B/N} \end{array}$$

The angular velocity error in the common frame is ${}^{B}\omega_{B/R}={\;}^{B}\omega_{B/N}-\delta C^{R}\omega_{R/N}$ and can be rewritten as ${}^{B}\omega_{B/N}={\;}^{B}\omega_{B/R}+\delta C^{R}\omega_{R/N}$: (48)$$\begin{array}{cc} \; & \dot{V}\left(\sigma_{B/R}\right)=\;(\sigma_{B/R}^{T}\;\left[-\frac{1}{N_{E}}\sum_{i=1}^{N_{E}}\ln\left(-\frac{C_{i}^{E}\left(\sigma_{B/N}\right)}{\alpha_{i}}\right)-\frac{1}{N_{I}}\sum_{j=1}^{N_{I}}\ln\left(\frac{C_{j}^{I}\left(\sigma_{B/N}\right)}{\beta_{j}}\right)\right]\\ \; & +2\ln\left(1+\sigma_{B/R}^{T}\sigma_{B/R}\right)\left[-\frac{1}{N_{E}}\sum_{i=1}^{N_{E}}\frac{{\left({[}^{B}\tilde{b}]{\left[BN\right]}^{N}\hat{n}\right)}^{T}}{C_{i}^{E}\left(\sigma_{B/N}\right)}-\frac{1}{N_{I}}\sum_{j=1}^{N_{I}}\frac{{\left({[}^{B}\tilde{b}]{\left[BN\right]}^{N}\hat{n}\right)}^{T}}{C_{j}^{I}\left(\sigma_{B/N}\right)}\right]){\;}^{B}\omega_{B/R}\\ \; & +2\ln\left(1+\sigma_{B/R}^{T}\sigma_{B/R}\right)\left[-\frac{1}{N_{E}}\sum_{i=1}^{N_{E}}\frac{{\left({[}^{B}\tilde{b}]{\left[BN\right]}^{N}\hat{n}\right)}^{T}}{C_{i}^{E}\left(\sigma_{B/N}\right)}-\frac{1}{N_{I}}\sum_{j=1}^{N_{I}}\frac{{\left(\left[{\;}^{B}\tilde{b}\right]{\left[BN\right]}^{N}\hat{n}\right)}^{T}}{C_{j}^{I}\left(\sigma_{B/N}\right)}\right]\delta C^{R}\omega_{R/N} \end{array}$$

Define $v_{T}$ and $u_{T}$ to be: (49)$$\begin{array}{cc} u_{T} & =2\ln\left(1+\sigma_{B/R}^{T}\sigma_{B/R}\right)\left[-\frac{1}{N_{E}}\sum_{i=1}^{N_{E}}\frac{{\left({[}^{B}\tilde{b}]{\left[BN\right]}^{N}\hat{n}\right)}^{T}}{C_{i}^{E}\left(\sigma_{B/N}\right)}-\frac{1}{N_{I}}\sum_{j=1}^{N_{I}}\frac{{\left({[}^{B}\tilde{b}]{\left[BN\right]}^{N}\hat{n}\right)}^{T}}{C_{j}^{I}\left(\sigma_{B/N}\right)}\right] \end{array}$$(50)$$\begin{array}{cc} v_{T} & =\sigma_{B/R}\;\left[-\frac{1}{N_{E}}\sum_{i=1}^{N_{E}}\ln\left(-\frac{C_{i}^{E}\left(\sigma_{B/N}\right)}{\alpha_{i}}\right)-\frac{1}{N_{I}}\sum_{j=1}^{N_{I}}\ln\left(\frac{C_{j}^{I}\left(\sigma_{B/N}\right)}{\beta_{j}}\right)\right]+u_{T} \end{array}$$

Then, Equation (Equation 46) is rewritten as:(51)$$\dot{V}\left(\sigma_{B/R}\right)=v_{T}^{TB}\omega_{B/R}+u_{T}^{T}\delta C^{R}\omega_{R/N}$$

The commanded angular velocity for this steering law is:(52)$${}^{B}\omega_{B^{*}/R}=-f\left(v_{T}\right)-\frac{v_{T}u_{T}^{T}}{v_{T}^{T}v_{T}}\delta C^{R}\omega_{R/N}$$
such that f satisfies the condition:(53)$$\dot{V}\left(\sigma_{B/R}\right)=-v_{T}^{T}f\left(v_{T}\right)\leq 0$$

A detailed derivation is conducted in Reference [32].

## 5. Results

The tracking performance is tested on a reference hill orbit frame with the following orbital parameters shown in Table 1. The reference frame R is built as: ${\hat{r}}_{1}$ is in the nadir direction, ${\hat{r}}_{2}$ is in the direction of the angular momentum, and ${\hat{r}}_{3}={\hat{r}}_{1}\times {\hat{r}}_{2}$. The spacecraft parameters and attitude pointing constraints are listed in Table 2. The spin projection matrix $\left[G_{s}\right]$ represents a four wheel RW array in a pyramid configuration with an interior angle of $55^{\circ }$.

A camera is placed in the x-axis of the body frame ${\hat{b}}_{1}={\;}^{B}x$ that must not enter the four exclusion zones. An antenna is placed in the y-axis of the body frame ${\hat{b}}_{2}={\;}^{B}y$ that must maintain pointing in the inclusion zone. MATLAB’s ode45 was used to propagate the states with a integration tolerance of $10^{-8}$. The initial conditions are an attitude of $\sigma_{0}={\left[\begin{array}{ccc} -0.67 & 0 & 0 \end{array}\right]}^{T}$, correlating to a $-135^{\circ }$ rotation about the x-axis, and angular velocity of $\omega ={\left[\begin{array}{ccc} 0 & 0 & \omega_{max} \end{array}\right]}^{T}$.

The common frame results are shown in Figure 9. The exclusion zones are shown as solid red circles, while the inclusion zones are shown as a dark green dotted circles. The simulation presents convergence to the target reference frame between 100 to 150 s without violating any pointing constraints. The MRPs switch to the shadow set at around 30 s into the maneuver. It is important to mention that the purpose of the presented work is to present the constrained attitude control formulation and simulation in the common frame dynamics. No attempt is made to tune the control parameters for better performance to arrive at a specific settling time for the problem.

The constraints represented by Equations (Equation 33) and (Equation 34), representing how close the boresight approaches the security angle of the constraint are plotted in the transient time in Figure 10 and Figure 11. It is shown that the first two exclusion constraints, $C_{1}$ and $C_{2}$ are more negative while the last two exclusion constraints $C_{3}$ and $C_{4}$ are less negative as an effect of Common Frame terms. In terms of the inclusion constraint $C_{5}$ are relatively more positive in the common frame.

### 5.1. Analysis of Standard and Common Frame Steering Law and Control Effort

The absolute error was taken from the original definition of the steering law and the new definition defined by Equation (Equation 52) for the entire maneuver. The transient time of the maneuver is shown in Figure 12, From Figure 12a, the Common Frame steering law requires relatively less angular rate compared to the original formulation as a result of the additional terms needed to remap the reference to the body frame and Figure 12b shows a relatively decreasing trend of the steering law effort. The Mean Absolute Error of the entire maneuver is about 1%.

The norm of the control effort was taken and the results are shown for the transient time of the maneuver in Figure 13. In Figure 13a, it is shown that the overall control effort is significantly reduced at about 7 seconds and that the common frame formulation converges faster than the original formulation as a result of additional terms presented in the servo subsystem as well as the steering law presented in Section 3. The average error throughout the entire maneuver is about 16%.

### 5.2. Monte Carlo Analysis

Figure 14 show the boresight trajectories given 100 runs with variance of given by the Table 3. The initial position of the boresights are marked as black circles and the end position is marked as a cyan x. The variance is multiplied by a random normal distribution given by the MATLAB function randn. Figure 15 shows the angle progression of boresight 1 with respect to the four exclusion constraints as well as the angle progression of boresight 2 with respect to the inclusion constraint, the dashed line is the minimum angle for each constraint from Table 2. In Figure 16, the histogram of the calculated constraints throughout the entire maneuver are shown for each constraint.

Another 100 Run Monte Carlo Simulation was conducted with the inclusion constraint removed. Comparing the trajectories of boresight 1 in Figure 17, the trajectories do not approach exclusion constraint 2 with the inclusion constrained removed.

In Figure 18, the angle progression between boresight 1 and the each exclusion constraint is shown and in Figure 19, it is shown that the minimum value for exclusion constraint 2 is more negative and exclusion constraint 4 is slightly less negative than in Figure 16.

### 5.3. Dynamic Constraint Performance

The proposed control design in Chapter 2 is tested on a combination of Type I (static constraints discussed previously) and Type III Constraints, or simply called dynamic conic constraints, where the position of the conic constraint varies with time. In this section, the same four static exclusion constraints are present along with a dynamic inclusion constraint. The reference frame is the same as previous sections from Table 1 and the same spacecraft parameters with exception to the inclusion constraint from Table 2 Consider the dynamic attitude inclusion pointing constraint: (54)$${}^{N}{\hat{n}}_{Dynamic},\theta_{min}={\left[0.93,\;\sin\left(\frac{\omega_{B/N_{z}}}{\gamma }t+\frac{\pi }{4}\right),\;0\right]}^{T}\;\;45^{\circ }$$
where $\omega_{B/Nz}$ is the z-component of the spacecraft current angular velocity, and γ is a scaling term for the angular velocity. In order to track the inclusion constraint, γ must be implemented such that the dynamic constraint does not move faster that the spacecraft in order to converge. The derivative of the dynamic inclusion constraint in Equation (Equation 54) is:(55)$${}^{N}{\dot{\hat{n}}}_{Dynamic}={\left[0,\;\frac{\omega_{B/N_{z}}}{\gamma }\cos\left(\frac{\omega_{B/N_{z}}}{\gamma }t+\frac{\pi }{4}\right),\;0\right]}^{T}$$

Figure 20 shows the simulation results of the constrained tracking maneuver under a dynamic inclusion constraint and four exclusion constraints under no torque bounds. The dynamic inclusion constraint seen in Figure 20f. In Figure 21, the inclusion constraint at different time steps are shown. For the dynamic constraints, the spacecraft reorients to the reference trajectory without violating any exclusion constraints as the dynamic constraint moves with time.

## 6. Conclusions

Constrained attitude control is a relatively new technology and is an active research topic with many proposed solutions. There are numerous advantages and disadvantages to using each solution. Geometric methods determine an intermediate waypoint outside the constraint in order to avoid the constraint. While geometric methods are relatively simple, these algorithms do not scale well when adding many constraints to the control problem. Methods using Constraint Monitor Algorithms is a real-time algorithm that actively monitors pointing constraints and creates a trajectory using a predictor-corrector approach, but convergence in the general case is not guaranteed. Optimal control methods can handle different types of constraints; however, these methods are usually complex in algorithm development. Lyapunov control methods have low complexity in implementation, but cannot solve the tracking problem alone and are not rate-bounded or torque bounded without modification. The main advantage of Lyapunov control is that the control laws are simple and ideal for real-time attitude control. In conjunction with using a backstepping control block that contains a steering law and a servo subsystem, the angular velocity rate can be bounded. Extending the conic constraints as a function of the maximum torque available for the reaction wheels array, a maximum moment of inertia, and maximum angular velocity bounds the control method in torque. This paper extends the benefits of this method by accommodating a common frame of reference for a scenario in which the true attitude is unknown as well as introducing dynamic constraints, where the constraint location is not an inertially fixed point and works well with static constraints. The main benefit of using common frame dynamics is that the result is more accurate to the maneuver than in the original constrained tracking problem.

## Figures and Tables

**Figure 1 sensors-22-10003-f001:**
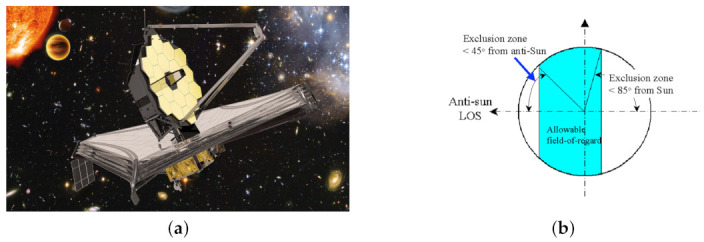
(**a**) The James Webb Space Telescope artist rendition. Source: European Space Agency. (**b**) JWST Field of Regard with two exclusion zones [3].

**Figure 2 sensors-22-10003-f002:**
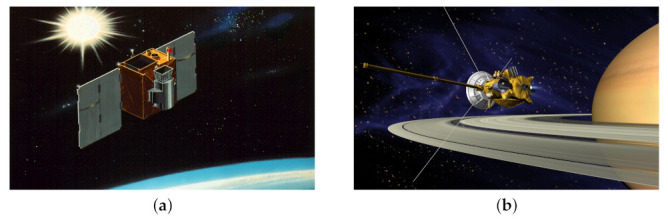
(**a**) SAMPEX Satellite, artist rendition (**b**) Cassini Satellite, artist rendition .

**Figure 3 sensors-22-10003-f003:**
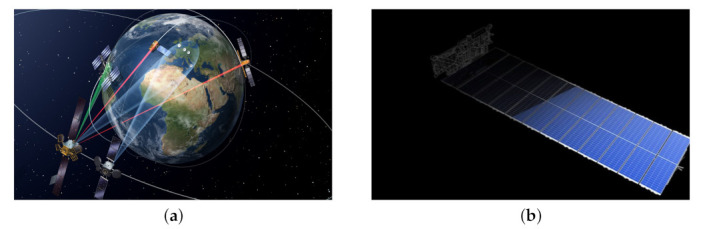
(**a**) European Data Relay System Satellite Constellation. Source: European Space Agency (**b**) One of many Starlink satellites, artist rendition. Source: SpaceX.

**Figure 4 sensors-22-10003-f004:**
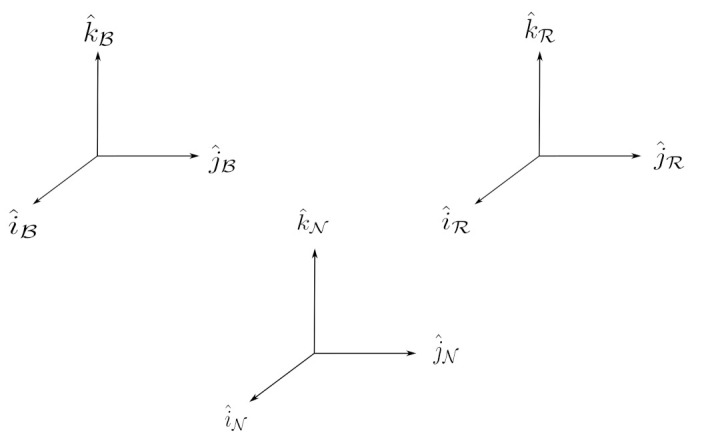
The relative coordinate system, with the inertially fixed earth axis N, reference trajectory R, and spacecraft body B.

**Figure 5 sensors-22-10003-f005:**
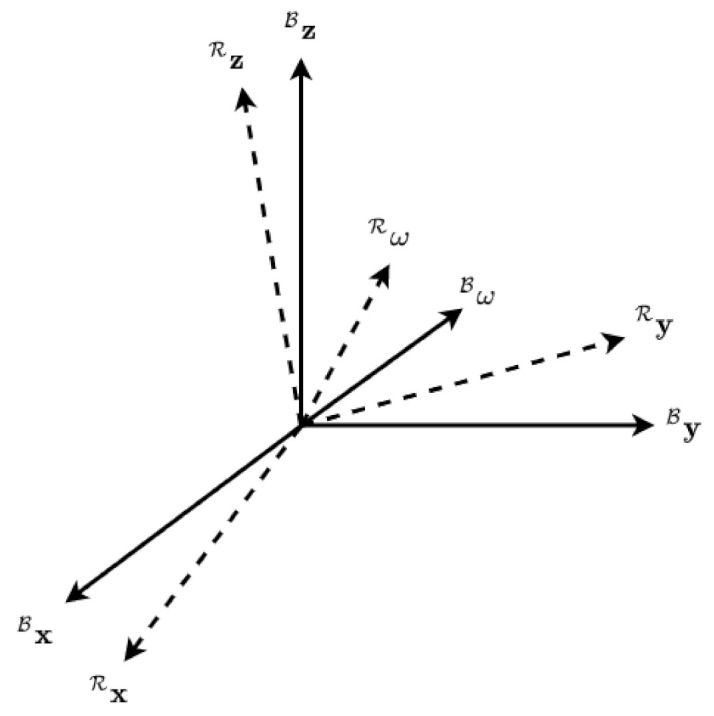
The body and reference frames and angular velocities shown in [24,25].

**Figure 6 sensors-22-10003-f006:**
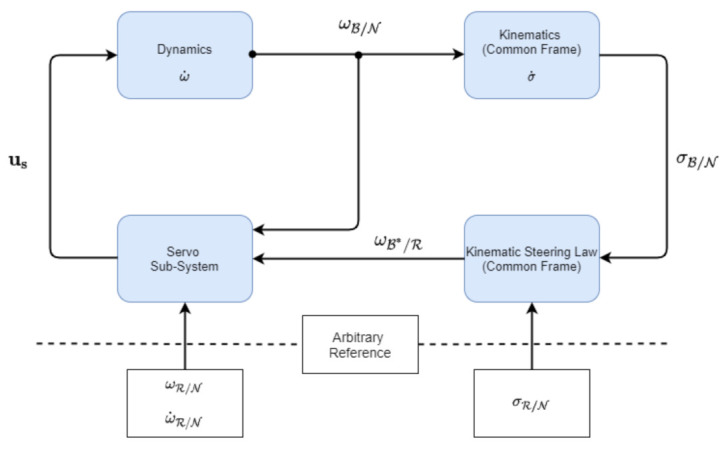
Control block proposed by [20]. The outer loop consists of the Kinematic Steering Law, while the inner loop consists of the servo sub-system.

**Figure 7 sensors-22-10003-f007:**
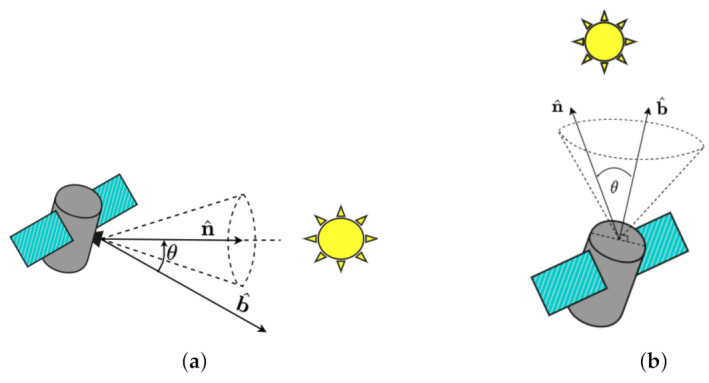
Static Pointing Constraints. The satellite in (**a**) has to keep its sensitive equipment from entering the exclusion cone defined by the sun while the satellite in (**b**) has an inclusion constraint defined by the sun while keeping its solar array pointed towards maximum power absorption.

**Figure 8 sensors-22-10003-f008:**
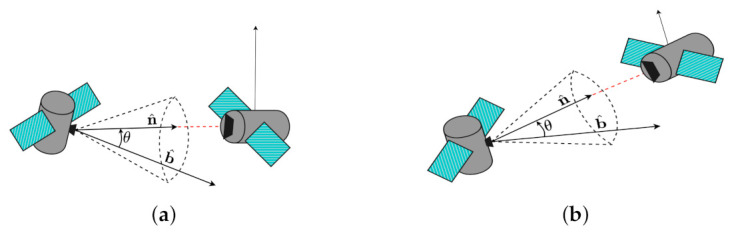
Dynamic Inclusion Constraint Geometry. A laser device is attached to a satellite on the left and must be pointed at another satellite’s receiver on the right. (**a**) is the initial position of the satellites and (**b**) is an arbitrary position after some specified time.

**Figure 9 sensors-22-10003-f009:**
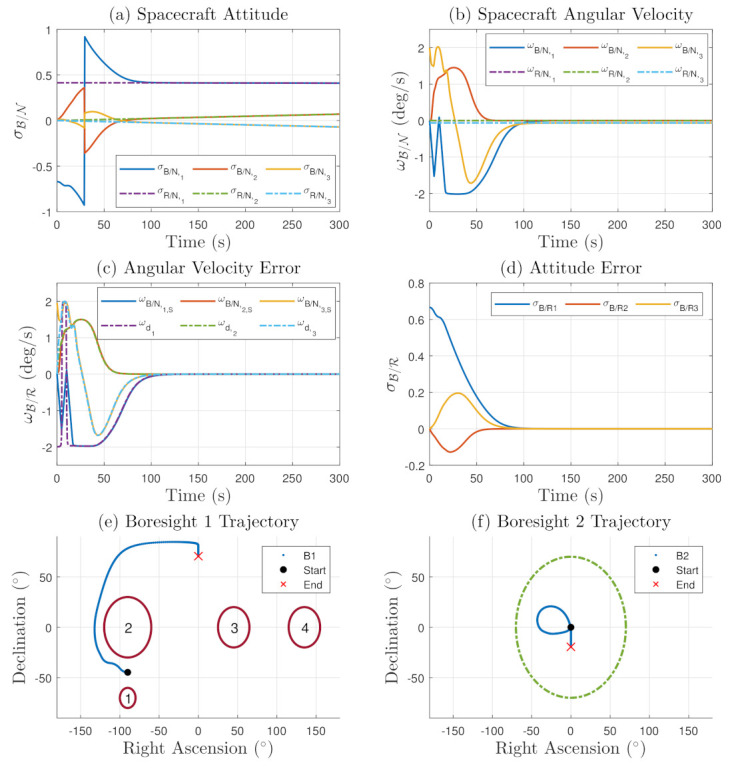
Unbounded Tracking Results in the Common Frame: (**a**) is the Spacecraft Attitude, (**b**) is the Angular velocity of the spacecraft, (**c**) is the Angular Velocity Error in the Common frame, (**d**) is the Attitude Error. (**e**, **f**) is are the boresight trajectories with respect to the constraints.

**Figure 10 sensors-22-10003-f010:**
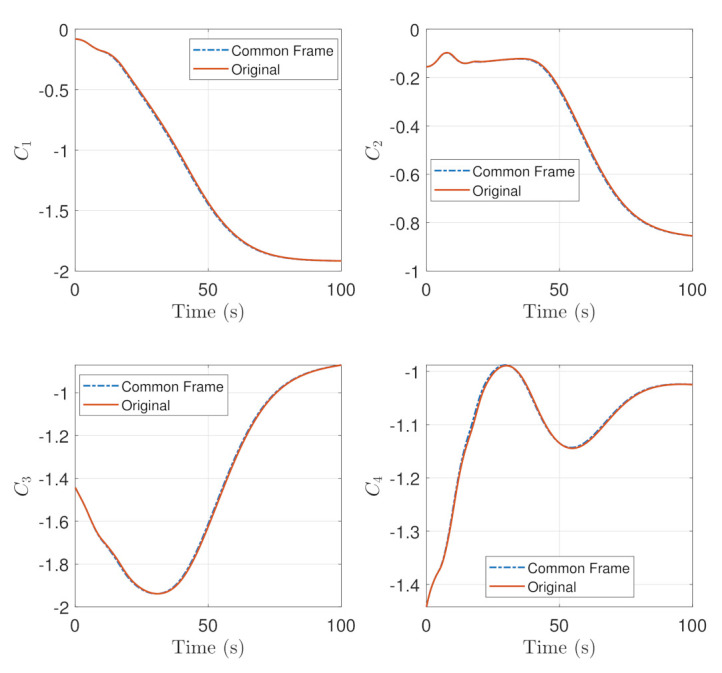
Comparison of Equation (Equation 33) in the Common Frame and the Original representation.

**Figure 11 sensors-22-10003-f011:**
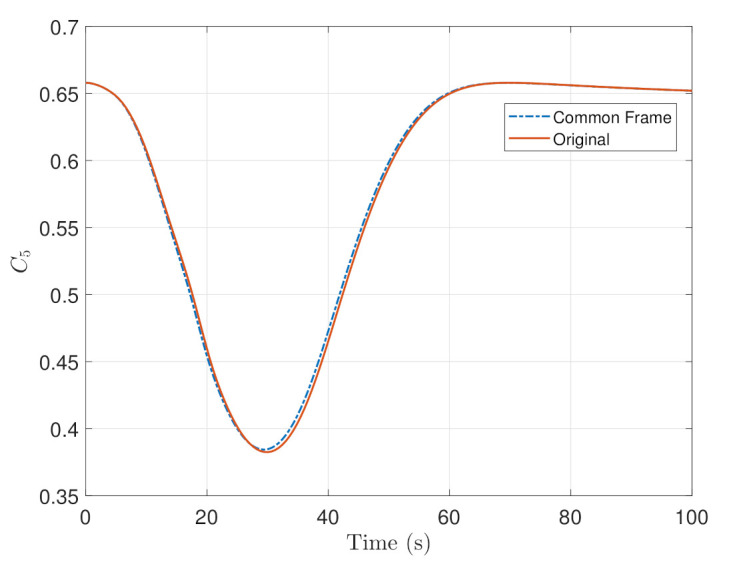
Comparison of Equation (Equation 34) in the Common Frame and the Original representation.

**Figure 12 sensors-22-10003-f012:**
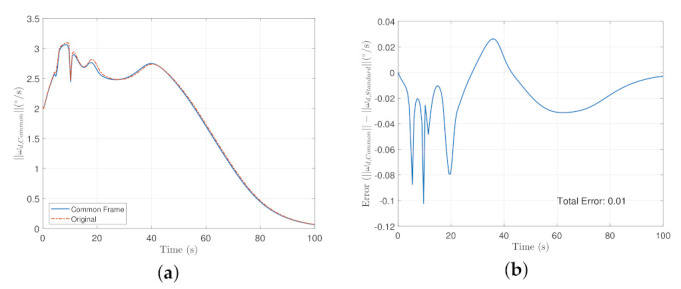
Transient time of the maneuver for the steering law where (**a**) is the overall commanded rates of the steering law, and (**b**) is the error plot of (**a**).

**Figure 13 sensors-22-10003-f013:**
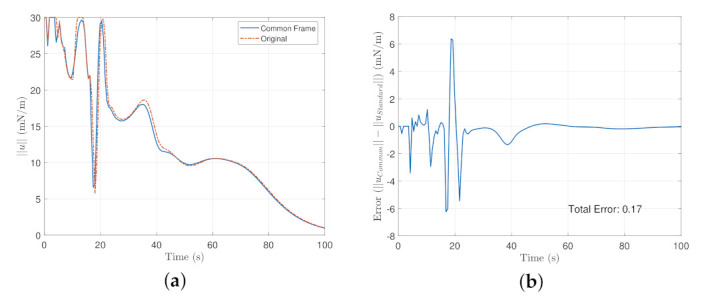
(**a**) is the Control Effort Norm in the Standard and the common frame in the transient time. while (**b**) is the error plot between the common frame and the original formulation.

**Figure 14 sensors-22-10003-f014:**
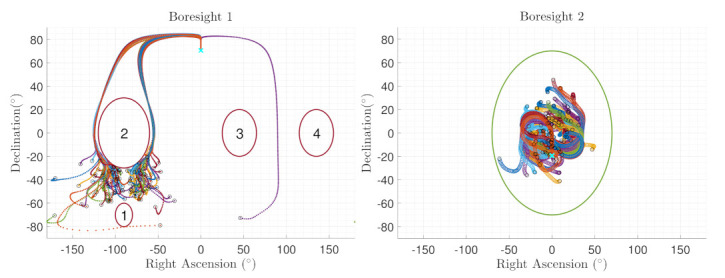
Monte Carlo results for sensitive equipment placed in the x-body axis with respect to the exclusion constraints (**Boresight 1**) and for equipment placed in the y-body axis with respect to the inclusion constraints (**Boresight 2**).

**Figure 15 sensors-22-10003-f015:**
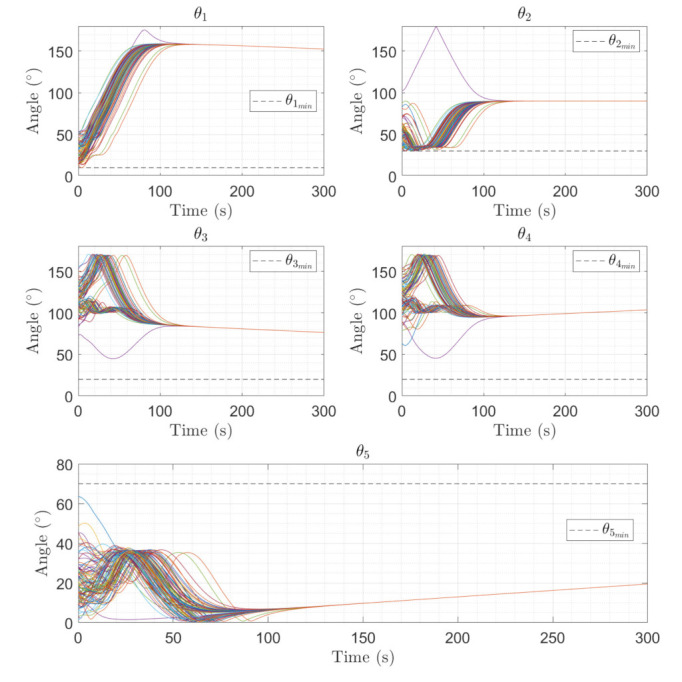
Angle progression of equipment with respect to the constraints. $\theta_{1}$ to $\theta_{4}$ represent the angle between Boresight 1 and the respective exclusion constraint, and $\theta_{5}$ is the angle between Boresight 2 with respect to the inclusion constraint.

**Figure 16 sensors-22-10003-f016:**
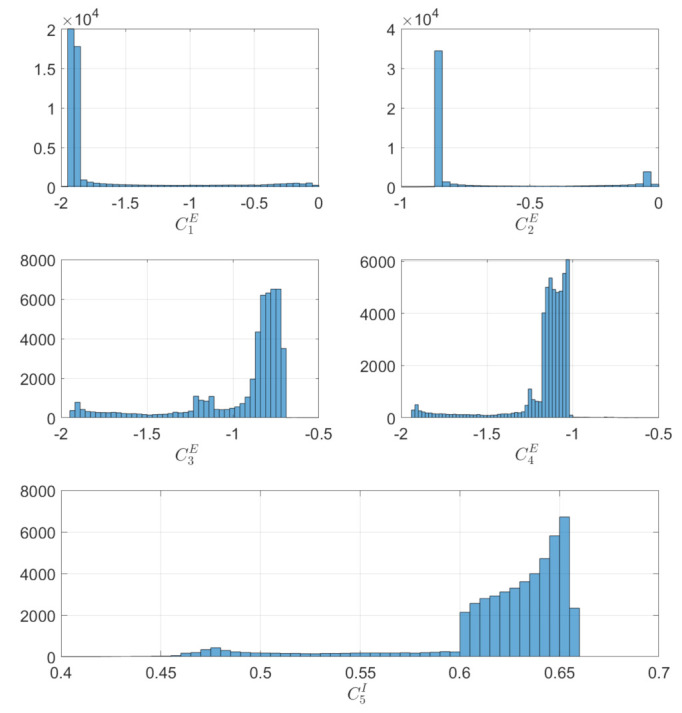
Histogram for the Constraint calculated in Equation (Equation 33) and Equation (Equation 34) under exclusion and inclusion constraints.

**Figure 17 sensors-22-10003-f017:**
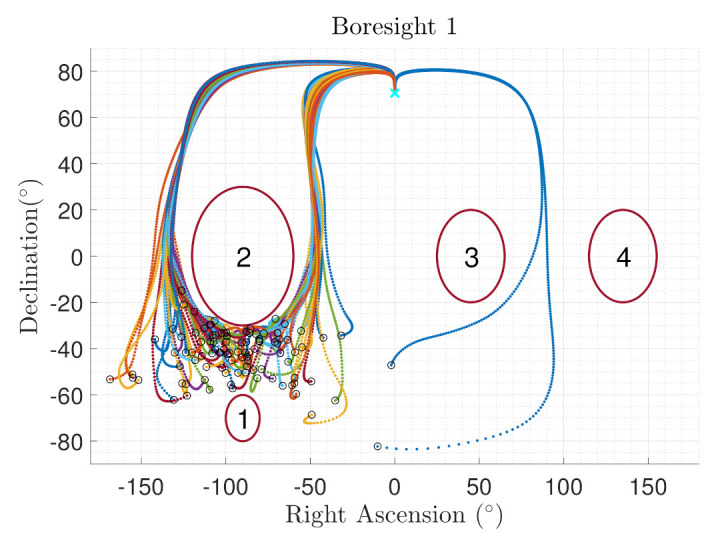
Monte Carlo results for sensitive equipment placed in the x-body axis (Boresight 1). The Inclusion Constraint is removed from the simulation.

**Figure 18 sensors-22-10003-f018:**
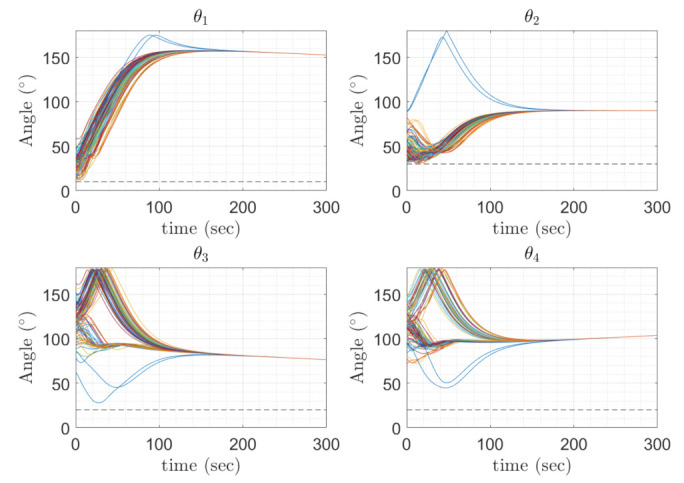
Boresight 1 trajectories with respect to the four exclusion constraints with the inclusion constraint condition removed.

**Figure 19 sensors-22-10003-f019:**
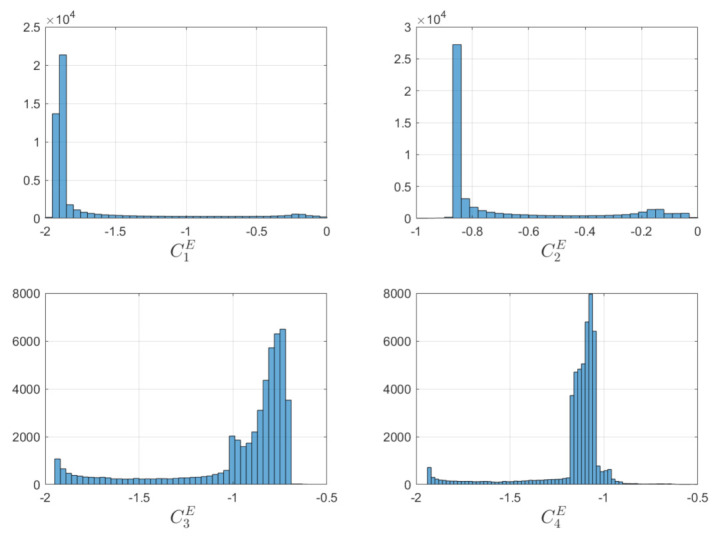
Histogram for the constraints calculated in Equation (Equation 33) purely under exclusion constraints.

**Figure 20 sensors-22-10003-f020:**
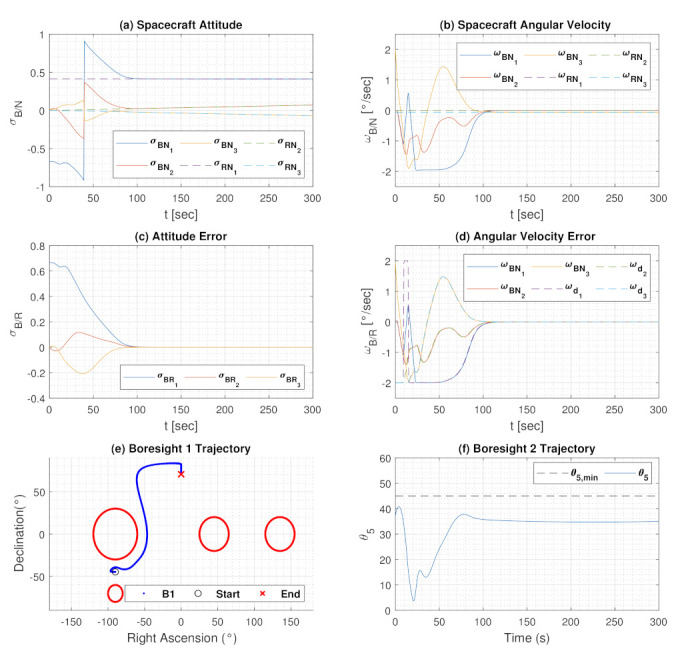
Tracking Performance with Dynamic Inclusion Constraint described by Equation (Equation 54). A γ=1.5 was used for the dynamic constraint, or half of $\omega_{z}$ current angular velocity. The spacecraft is unbounded in torque. (**a**) is the spacecraft attitude history, (**b**) is the angular velocity history, (**c**) is the attitude error, (**d**) is the angular velocity error, (**e**) is the trajectory of boresight 1, and (**f**) is the angle history of boresight 2.

**Figure 21 sensors-22-10003-f021:**
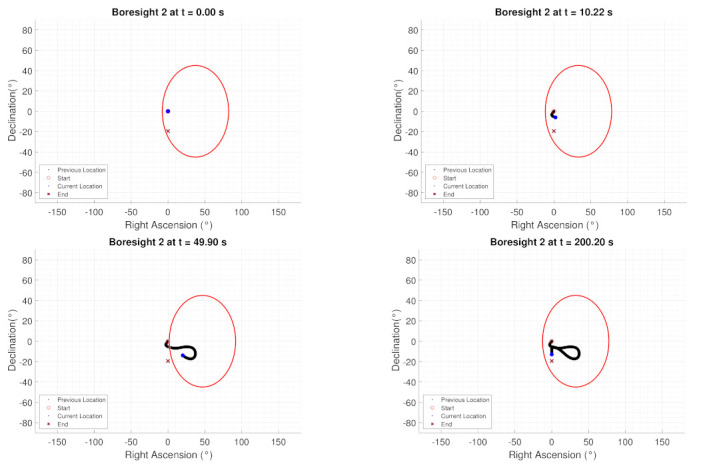
Boresight 2 trajectory at different times of the simulation, γ=1.5 is used.

**Table 1 sensors-22-10003-t001:** Orbital Parameters for Reference Frame, Values obtained from [20].

Orbital Parameter	Value
Earth Radius	6378.0 km
Earth Gravitational Parameter	398,600.00 km${}^{3}/$s${}^{2}$
Right Ascension of ascending node	$0^{\circ }$
Inclination	$-90^{\circ }$
Orbital Altitude	400 km
Initial argument of Latitude	$180^{\circ }$

**Table 2 sensors-22-10003-t002:** Spacecraft and Pointing Constraint Parameters, Values obtained from [20] .

Description	Variable	Value
Spacecraft Inertia Tensor	$\left[I_{S}\right]$	diag$\left(\left[\begin{array}{ccc} 4.415 & 4.415 & 3.83 \end{array}\right]\right)$ km · m${}^{2}$
Max Angular Velocity	$\omega_{max}$	2 ${}^{\circ }$/s
RW Parameters	$\left[I_{W}\right]$	diag$\left(\left[\begin{array}{ccc} 0.03 & 0.001 & 0.001 \end{array}\right]\right)$ km · m${}^{2}$
	$\left[G_{s}\right]$	$\left[\begin{array}{cccc} 0.819 & 0 & -0.819 & 0\\ 0 & 0.819 & 0 & -0.819\\ 0.5736 & 0.5736 & 0.5736 & 0.5736 \end{array}\right]$
	$u_{s_{max}}$	15 mN · m
	Ω	$500{\left[\begin{array}{cccc} 1 & -1 & 1 & -1 \end{array}\right]}^{T}$ rpm
Control Gains	[P]	10$\left[I_{3\times 3}\right]$
	$\left[K_{I}\right]$	0.01$\left[I_{3\times 3}\right]$
Smoothing Constants	$\left[K_{1}\right],\left[K_{3}\right]$	0.1
Exclusion Constraints	${}^{N}{\hat{n}}_{1}$,$\theta_{min_{1}}$	${\left[\begin{array}{ccc} 0 & -0.34 & -0.96 \end{array}\right]}^{T}$, $10^{\circ }$
	${}^{N}{\hat{n}}_{2}$,$\theta_{min_{2}}$	${\left[\begin{array}{ccc} 0 & -1 & -0.96 \end{array}\right]}^{T}$, $30^{\circ }$
	${}^{N}{\hat{n}}_{3}$,$\theta_{min_{3}}$	${\left[\begin{array}{ccc} 1 & 1 & 0 \end{array}\right]}^{T}$, $20^{\circ }$
	${}^{N}{\hat{n}}_{4}$,$\theta_{min_{4}}$	${\left[\begin{array}{ccc} -1 & 1 & 0 \end{array}\right]}^{T}$, $20^{\circ }$
Inclusion Constraints	${}^{N}{\hat{n}}_{5}$,$\theta_{min_{5}}$	${\left[\begin{array}{ccc} 1 & 0 & 0 \end{array}\right]}^{T}$, $70^{\circ }$
Moving Average Window	$f_{outer}$	0.5 s

**Table 3 sensors-22-10003-t003:** Variance Parameters for the Initial Attitude ($\sigma_{B/N_{0}}$) and Initial Angular Velocity ($\omega_{B/N_{0}}$) .

Description	Value
Attitude Variance	0.1
Angular Velocity Variance	0.1

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
