# Peer review of "Common Frame Dynamics for Conically-Constrained Spacecraft Attitude Control"

_sensors, 2022, doi:10.3390/s222410003_

Round 1
Reviewer 1 Report
Common Frame Dynamics for Conically-Constrained Spacecraft Attitude Control
1. The main goal of this paper is: Attitude control for constrained spacecraft by solving the constrained attitude control problem using common frame dynamics
2. The initial part - Introduction:
2.1. The literature review must be improved. You need to improve the text so that the reader can better understand what you want to say. In some cases, it is difficult to understand what the authors of the papers did;
2.2. (line 36) You stated "the relevant literature to the constrained attitude control problem is sparse". That's true? Have you looked through all the research sources to say it is sparse? You must be careful when stating this;
2.3. (lines 73, 78 and 113) You use acronyms such as “NLP, SDP and MRP” but don't say what they mean. I know they are common acronyms when it comes to control but it is important that everything is indicated throughout the text;
3. The dynamic and control part:
3.1. I think it would be interesting you explain in a little more detail how to obtain the constraint equations;
3.2. For example, in lines 182 and 183 you do not detail the constraints;
4. Results:
4.1. In table 2 you inform that the values were obtained in [20] but I think it is important that you inform the details of your structure here and don't leave it indicated. Inform which structure is this analyzed. Is it a real structure or just numerical?;
4.2. In figure 9, it is important to inform in the figure itself, which are the references;
4.3. Figure 9 shows that its control system takes approximately 100 seconds to reach the target. This time is not too high? In 100 seconds, a lot can happen to your system, especially at high speeds;
4.4. How did you get the control gains in Table 2? This is one of the most important information in a control system. Have you tested the control for other gain values? Are these optimal gains?;
4.5. I think it would be interesting to plot the error, Figures 10 and 11, of each constraint considering the two approaches;
4.6. Is the reference in figures 12 and 13 the same as in the previous figures? It is necessary to put the reference maneuver for the two techniques in order to visualize how the controller behaves;
4.7. The convergence time of the curves, in Figures 15 and 18, is greater than 100 seconds. Is this value too high? Please explain more about this value;
4.8. Where does equation 47 come from?;
4.9. In figure 20 there is an error. Figure (c) is not the attitude error;
4.10. The Figure 20(f) don´t exist;
4.11. You stated, in line 245 and 246, "the spacecraft reorients to the reference trajectory without violating any exclusion constraints as the dynamic constraint moves with time" but in Figure 21 is there a constraint violation or not? Explain the third figure better.
5. Conclusion:
5.1. You use acronyms in conclusions. I suggest not using them;
5.2. You stated that "It can be shown that the maneuver in both unbounded torque and the bounded torque problem has an overall control effort error of about 16% in the unbounded torque scenario" but it is not clear in the text this 16% error. Where are the limited torque simulations? They need to be clear to the reader;
5.3. To validate your technique, it would be interesting to add noise to the measurements to see if your controller is still able to control the system satisfactorily. It is to verify the robustness of the methodology;
5.4. At no time is it very clear what the contribution of your work is. What makes it different from other works in the same area?
5.5. Is the computational cost to control the system is high or lower? This was not stated in the text;
5.6. Is your technique viable to be implemented experimentally? What difficulties and facilities would you have to implement it experimentally?
Reviewer 2 Report
Journal Review: Physical Sensors
Overview:
Thank you for requesting my review on this amazing work. The paper addresses some key practical concerns that spacecraft GNC engineers face in satellite operations. I highly commend the rigorous derivation and the visualizations that went into the results. Overall, I believe this paper would make for a great publication in this Journal. Below are some suggestions that, in my humble opinion, can better underscore the contributions of the current work, and thereby improve the accessibility to readers that might not readily have the background expected in the current work.
Major Comments
-
While the key focus of the paper is on describing the dynamics, which has been done in a rigorous and stellar fashion, the paper seems to be limited in explaining the motivation for such a reformulation. The main suggestion here is to include a separate “Motivation” section to describe the need for such a dynamical reformulation, which should also be reflected in the abstract. The following suggestions may help provide more context, and also better underscore the contributions of this work:
-
Can the problem of conical constraints be solved as a simple reference trajectory tracking problem?- Here the user would force the state to track a reference trajectory. The paper does mention references to Sliding/ adaptive control laws that solve the problem this way. But does not address the advantages/ limitations of the current method. Questions to address here are as follows:
-
Can a reference tracking controller solve the current problem with conical constraints?
-
If yes, then what benefit do we get from switching to the new dynamical formulation?
-
If not, then why do those methods fail?
-
Also to clarify, a full dynamical simulation between existing methods is not being requested here, rather the request is to provide the expected advantages and disadvantages of the current method over the state-of-the-art.
-
Based solely on personal interpretation, what is being referred to as “Common Frame Dynamics” is generally called Attitude Kinematics. A suggestion here is to clarify if that is or is not the case, early on in Section 1.
-
The authors selected Modified Rodrigues Parameter to represent the attitude coordinates. Some clarification on this decision can be beneficial to the readers. Specifically:
-
Did the authors get any benefit from formulating the kinematics in MRPs, or can the same benefits be obtained from DCMs and quaternions?
-
To clarify, the request here is not a new formulation, rather comment on why the MRP formulation was used. It is completely valid to say something along the lines of “MRPs were selected because the authors are more familiar with them.” However making this decision explicit will shed light on alternate formulations.
Minor Comments
-
Since the objective of the paper is to steer the angle, theta, along a certain constrained trajectory, it might be helpful to the readers to understand which of the terms in the controller/ derivation are effected by theta.
-
Request/ suggestion here is to explicitly indicate which parameters in Section 4.4 depend on theta. Ie if parameter x depends on theta, label it as x(theta).
-
Minor rephrasing of the Abstract might help with better (typed in blue below):
-
Pointing constraints can be divided into two categories: Exclusion zones that are defined for sensitive equipment such as telescopes or cameras that can be damaged from celestial objects; and Inclusion zones that are defined for communication hardware and solar arrays.
-
This work derives common frame dynamics that are fully derived for Modified Rodrigues Parameters and introduced to an existing novel technique for constrained spacecraft attitude control, which uses a kinematic steering law and servo sub-system.
-
Author Credentials on Page 1 are missing.

Author Response
See attached file, please.

Round 2
Reviewer 1 Report
Common Frame Dynamics for Conically-Constrained Spacecraft Attitude Control
1. The main goal of this paper is: Attitude control for constrained spacecraft by solving the constrained attitude control problem using common frame dynamics
2. Results:
2.1. In your answer you state that: “4.3 The main purpose of the presented work is to present the constrained attitude control formulation and simulation in the common frame dynamics. We did not attempt tuning the control parameters for better performance. The gains are not tuned to arrive in specific settling time for the problem. In the future work, we will consider tuning the gain parameters to meet specific control performance such as settling time and system overshoot”. I think it's important that you inform this in the text;
2.2. In your answer you state that: “4.6 Again, we did not attempt to select or tune the control gains for better performance. The example is shown to demonstrate the common frame constrained problem for a given control setup. Monte Carlo is conducted for validation purposes. In the future work, we will consider tuning the gain parameters to meet specific control performance”. I think it's important that you inform this in the text;
3. Conclusion:
3.1. You continue using acronyms in conclusions. I suggest not using them.
